# N-acetylation of α-synuclein enhances synaptic vesicle clustering mediated by α-synuclein and lysophosphatidylcholine

**Chuchu Wang**[1,2,3†]**, Chunyu Zhao**[1,2†]**, Hu Xiao**[4]**, Jiali Qiang**[1,2]**, Zhenying Liu**[1,2]**, Jinge Gu**[1,2]**, Shengnan Zhang**[1,2]**, Dan Li**[5,6]**, Yaoyang Zhang**[1]**, Jacqueline Burré**[7]**, Jiajia Diao**[4]*****, Cong Liu**[1,8]*****

[1]Interdisciplinary Research Center on Biology and Chemistry, Shanghai Institute of Organic Chemistry, Chinese Academy of Sciences, Shanghai, China; [2]University of Chinese Academy of Sciences, Beijing, China; [3]Department of Molecular and Cellular Physiology, Stanford University, Stanford, United States; [4]Department of Cancer Biology, University of Cincinnati College of Medicine, Cincinnati, United States; [5]Bio-X Institutes, Key Laboratory for the Genetics of Developmental and Neuropsychiatric Disorders (Ministry of Education), Shanghai Jiao Tong University, Shanghai, China; [6]Zhangjiang Institute for Advanced Study, Shanghai Jiao Tong University, Shanghai, China; [7]Brain and Mind Research Institute & Appel Institute for Alzheimer's Disease Research, Weill Cornell Medicine, New York, United States; [8]State Key Laboratory of Chemical Biology, Shanghai Institute of Organic Chemistry, Chinese Academy of Sciences, Shanghai, China

*For correspondence:
diaoje@ucmail.uc.edu (JD);
liulab@sioc.ac.cn (CL)

†These authors contributed equally to this work

Competing interest: The authors declare that no competing interests exist.

## eLife assessment

In this **useful** study, the authors show that N-acetylation of synuclein increases clustering of synaptic vesicles in vitro and that this effect is mediated by enhanced interaction with lysophosphatidylcholine. While the evidence for enhanced clustering is largely **solid**, the biological significance remains unclear.

**Abstract** Previously, we reported that α-synuclein (α-syn) clusters synaptic vesicles (SV) Diao et al., 2013, and neutral phospholipid lysophosphatidylcholine (LPC) can mediate this clustering Lai et al., 2023. Meanwhile, post-translational modifications (PTMs) of α-syn such as acetylation and phosphorylation play important yet distinct roles in regulating α-syn conformation, membrane binding, and amyloid aggregation. However, how PTMs regulate α-syn function in presynaptic terminals remains unclear. Here, based on our previous findings, we further demonstrate that N-terminal acetylation, which occurs under physiological conditions and is irreversible in mammalian cells, significantly enhances the functional activity of α-syn in clustering SVs. Mechanistic studies reveal that this enhancement is caused by the N-acetylation-promoted insertion of α-syn's N-terminus and increased intermolecular interactions on the LPC-containing membrane. N-acetylation in our work is shown to fine-tune the interaction between α-syn and LPC, mediating α-syn's role in synaptic vesicle clustering.

## Introduction

α-Syn is a highly abundant presynaptic protein in neurons. It is involved in synaptic vesicle (SV) clustering and the assembly of soluble N-ethylmaleimide sensitive factor receptor (SNARE) complex for mediating SV trafficking and neurotransmitter release (*Diao et al., 2013*; *Burré et al., 2010*; *Wang et al., 2024*). Abnormal amyloid aggregation of α-syn and deposition into Lewy Bodies (LBs) is a pathological hallmark of Parkinson's disease (*Goedert et al., 2013*; *Uversky and Eliezer, 2009*; *Li and Liu, 2022*). Different types of PTMs, e.g., acetylation, ubiquitination, and phosphorylation have been identified to modify α-syn under physiological and pathological conditions (*Uversky and Eliezer, 2009*; *Fauvet et al., 2012*; *Oueslati, 2016*; *Mahul-Mellier et al., 2014*; *Zhang et al., 2019*; *Hu et al., 2024*). Phosphorylation of S129 and Y39 is elevated, and this PTM-modified α-syn accumulates in LBs, implying a direct relationship between PTMs and α-syn pathology (*Oueslati, 2016*; *Mahul-Mellier et al., 2014*). Rather than disease-related PTMs, additional PTMs e.g., N-acetylation and O-GlcNAcylation of α-syn were found also under normal conditions (*Theillet et al., 2016*; *Marotta et al., 2015*; *Li et al., 2023*). Specifically, N-acetylation can influence the conformation and increase the helicity of the N-terminal region of α-syn, alters its membrane-binding behavior, and is critical for oligomer formation and amyloid aggregation kinetics (*Bu et al., 2017*; *Runfola et al., 2020*; *Trexler and Rhoades, 2012*). However, it remains unknown whether N-acetylation directly regulates the physiological function of α-syn in clustering SVs.

## Results

### N-terminal acetylation enhances SV clustering induced by α-syn

In this study, we sought to investigate whether N-acetylation could modulate α-syn's function in promoting SV clustering. Firstly, we prepared recombinant N-terminally acetylated α-syn (Ac-α-syn) as well as unmodified α-syn (un-α-syn). N-terminal acetylation leads to a significant chemical shift perturbation of the first nine residues and H50 of α-syn in the solution nuclear magnetic resonance (NMR) spectrum (*Figure 1a*, left panel). Intriguingly, in-cell NMR experiments show that all of the un-α-syn is N-terminally acetylated once it is delivered into HEK-293T cells by electroporation (*Figure 1a*, right panel), which is consistent with a previous report (*Theillet et al., 2016*). This result confirms the physiological relevance of N-terminal acetylation of α-syn. Notably, N-terminal acetylation is believed to be irreversible in mammalian cells due to the lack of a known N-terminal deacetyltransferase. Hence, it is crucial to investigate the impact of N-acetylation on the function of α-syn. To further probe the influence of N-acetylation on α-syn-mediated SV clustering, we incubated mouse SVs with various α-syn variants (*Figure 1b*, upper panel). We next measured the extent of SV clustering by utilizing dynamic light scattering (DLS) to monitor alterations in particle diameters, as well as using negatively stained transmission electron microscopy (TEM) (*Figure 1b*, middle and lower panel). Notably, N-acetylation significantly amplified α-syn's capacity to cluster SVs, which requires the N-terminal thirty residues (*Figure 1c–d*). These findings imply that N-acetylation augments SV clustering.

### LPC mediates the enhancement of vesicle clustering by N-terminal acetylation of α-syn

In our previous work, we found that α-syn can bind to both negatively charged dioleoyl-phosphoserine (DOPS) (*Diao et al., 2013*) and neutral LPC (*Lai et al., 2023*). Building on this, we sought to examine how N-acetylation affects SV clustering that arises from the interaction between α-syn and either of these lipids. To assess clustering functionality with specific lipid compositions, we synthesized liposomes mimicking SVs that contained either LPC or DOPS. We then used a single-vesicle clustering assay to track the clustering mediated by Ac-α-syn and un-α-syn (*Figure 2a*). Importantly, we observed that N-acetylation considerably boosted the clustering ability of α-syn with LPC-containing liposomes (*Figure 2b*). In contrast, N-acetylation did not increase the clustering of liposomes containing DOPS (*Figure 2c*). Therefore, LPC is responsible for the increased SV clustering activity by N-terminally acetylated α-syn.

### N-terminal acetylation increases the α-syn–LPC interaction

The interaction between α-syn and lipids is crucial for clustering SVs (*Diao et al., 2013*; *Lai et al., 2023*; *Fusco et al., 2016*). To elucidate the structural basis of how N-acetylation impacts the α-syn–LPC

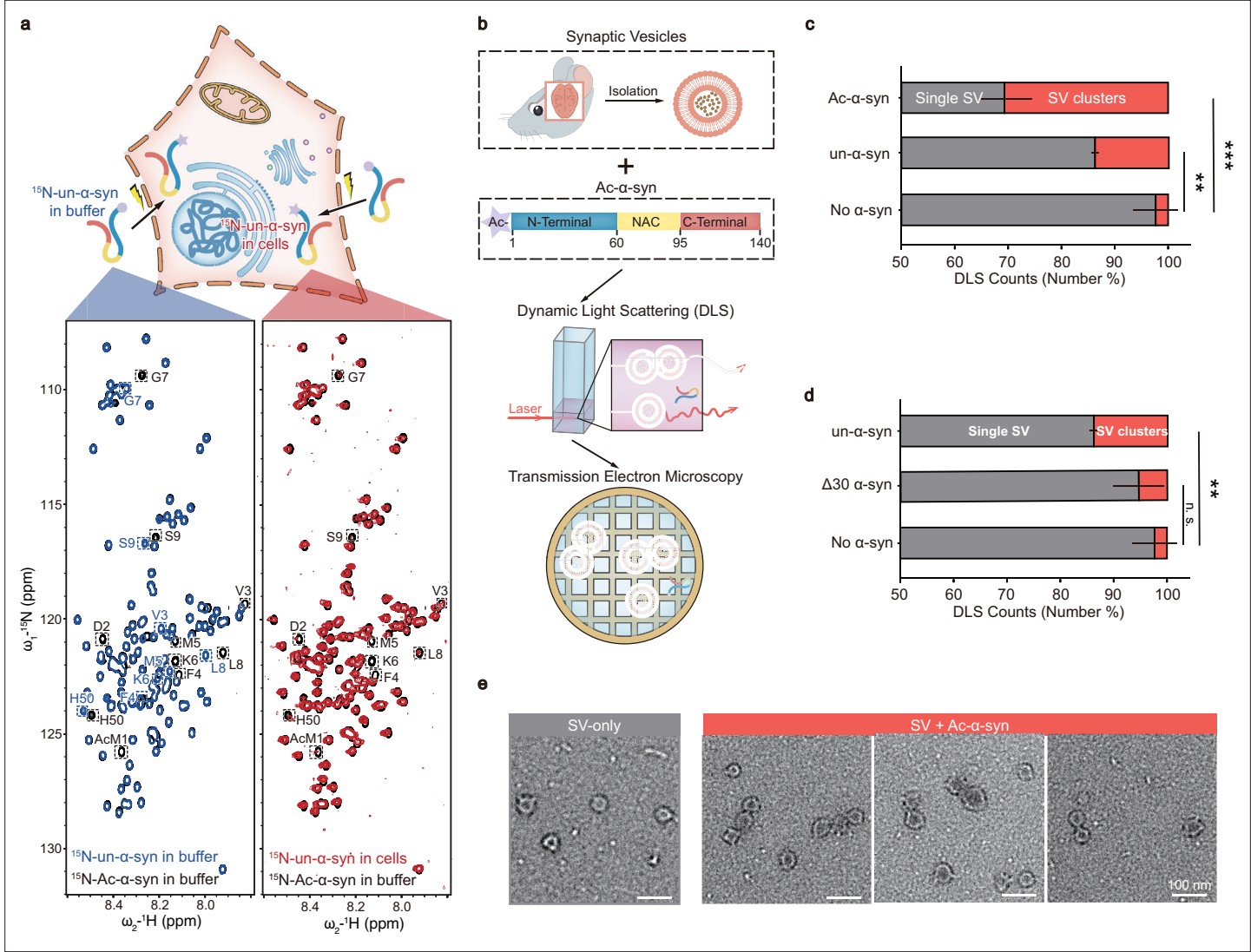

**Figure 1.** N-terminal acetylation enhances synaptic vesicle (SV) clustering induced by α-synuclein (α-syn). (**a**) α-syn is N-terminally acetylated in cells. Upper panel: Schematic representation of the delivery of ¹⁵N-labeled N-terminally unmodified α-syn (un-α-syn) into mammalian cells through electroporation. Lower panel: Comparisons of 2D ¹H-¹⁵N HSQC spectra of N-acetylated α-syn (Ac-α-syn) (black) with in-buffer un-α-syn (blue) and in-cell un-α-syn (red). Distinct assignments for nuclear magnetic resonance (NMR) cross-peaks corresponding to amino acids in Ac-α-syn and un-α-syn are enclosed and labeled. (**b**) Scheme of experimental design for the functional study of Ac-α-syn on SV clustering. SVs isolated from mouse brains were added to Ac-α-syn (Upper) for measuring size distribution by dynamic light scattering (DLS) (middle panel), and the sample was further visualized by using negatively stained transmission electron microscopy (TEM) (lower panel). (**c**) The influences of N-acetylation of α-syn and (**d**) α-syn without the N-terminal thirty residues on SV clustering measured by DLS. The X-axis represents the number percent of the single SV and clustered SVs counted by DLS. Error bars are standard deviations from three biological replicates. **p-value <0.01; ***p-value <0.001; analysis by Student's t-test. (**e**) Representative negatively stained TEM images of the single SV and clustered SVs in SV samples with no α-syn (gray) and Ac-α-syn (red), respectively. *Figure 1—source data 1* (c&d): the DLS numerical data of SV&Ac-α-syn, SV&un-α-syn, SV&Δ30-α-Syn, and SV-only (three biological replicates of each sample).

The online version of this article includes the following source data for figure 1:

**Source data 1.** The DLS numerical data.

interaction, we carried out solution NMR titration analyses to examine the α-syn−LPC interaction with residue-specific resolution. Adding increasing concentrations of pure LPC micelles to unmodified α-syn resulted in a dose-dependent and continuous attenuation of signals for the first 100 residues of α-syn, indicating that these residues can interact with LPC (*Figure 3a*, upper). Notably, N-acetylation significantly elevated the binding affinity of the N-terminal region (residues 1–30), particularly the initial 10 residues, towards LPC, leading to a rapid binding pattern during titration (*Figure 3a*,

**Figure 2.** Lysophosphatidylcholine (LPC) mediates the enhancement of vesicle clustering by N-terminal acetylation of α-synuclein (α-syn). (**a**) Scheme of single-vesicle clustering assay for the functional study of Ac-α-syn on vesicle clustering. Vesicles were prepared with different amounts of LPC, and were labeled with DiD or remained unlabeled, respectively. A saturated layer of unlabeled vesicles was immobilized on the imaging surface. Free DiD-vesicles were injected into the system with Ac-α-syn. Red laser illumination imaged the DiD-vesicles that clustered with unlabeled vesicles. The enhancement of LPC (**b**) and dioleoyl-phosphoserine (DOPS) (**c**) on single vesicle clustering count by Ac-α-syn and un-α-syn, respectively, was measured. Error bars are standard deviations from six random imaging locations in the same sample channel. *** indicates p-value <0.001, analysis by Student's t-test.

lower). Moreover, Ac-α-syn demonstrated a sequential binding behavior with LPC, where the N-terminal region (residues 1–30) bound initially, succeeded by the central region (residues 30–100) as the LPC/Ac-α-syn ratio increased (*Figure 3a*, lower). Similar trends were observed in NMR titrations using DOPC/LPC-mixed liposomes (*Figure 3b*). On the other hand, N-acetylation only slightly weakened the interaction between the N-terminal region (residues 1–30) and DOPS, particularly at low DOPS/α-syn ratios, resulting in a continuous signal attenuation for the first 100 residues (*Figure 3c*). These findings reveal that N-acetylation considerably enhances the N-terminal region's binding affinity, particularly the first 10 residues, to LPC, while modestly reducing the same region's affinity to DOPS.

To provide insights that are more relevant to physiological conditions, we extended our investigation to examine how Ac-α-syn interacts with SV membranes. By utilizing a protocol established before (*Wang et al., 2020*), we prepared monodisperse SVs isolated from normal mouse brains and titrated them with Ac-α-syn at an SV/α-syn ratio of 260:3000, which is a physiologically relevant ratio within presynaptic terminals as determined in previous work (*Wilhelm et al., 2014*). The HSQC spectrum revealed signal attenuation, notably in the N-terminal region and specifically within the first ten residues of α-syn (*Figure 3d*). These findings closely mirrored those observed when Ac-α-syn was titrated with LPC micelles and DOPC/LPC-mixed liposomes at low lipid/α-syn ratios of 10:1 and 100:1, respectively (*Figure 3e*). These results further substantiate the critical role played by the N-terminal region of Ac-α-syn in SV membrane binding.

## Ac-α-syn binding on LPC shows high intermolecular interactions

The intermolecular interactions between α-syn monomers (*Fusco et al., 2016*) or between α-syn and vesicle-associated membrane protein 2 (VAMP2) (*Diao et al., 2013*) are also essential for clustering SVs. To further characterize the accessibility of α-syn to LPC and DOPS, we performed cross-linking experiments coupled with mass spectrometry (XL-MS). We mixed $^{15}$N-labeled and unlabeled Ac-α-syn at a 1:1 ratio and selected the cross-linked fragments containing both $^{15}$N-labeled and unlabeled Ac-α-syn to map the intermolecular cross-linking pattern. We found the accessibility of Ac-α-syn to LPC to be much higher than that to DOPS (*Figure 4a*). The number of cross-linked peptide pairs, indicative of their close interaction, detected in LPC-induced multimers was over two-fold more than that of DOPS (*Supplementary file 1a-f*). This result demonstrates high intermolecular interactions between Ac-α-syn and LPC.

## Discussion

In summary, the interaction between α-syn and lipids plays a crucial role in both the protein's physiological functions in SV trafficking and its pathological aggregation, as observed in neurodegenerative conditions (*Burré et al., 2018*). Understanding how PTMs like N-acetylation modulate these

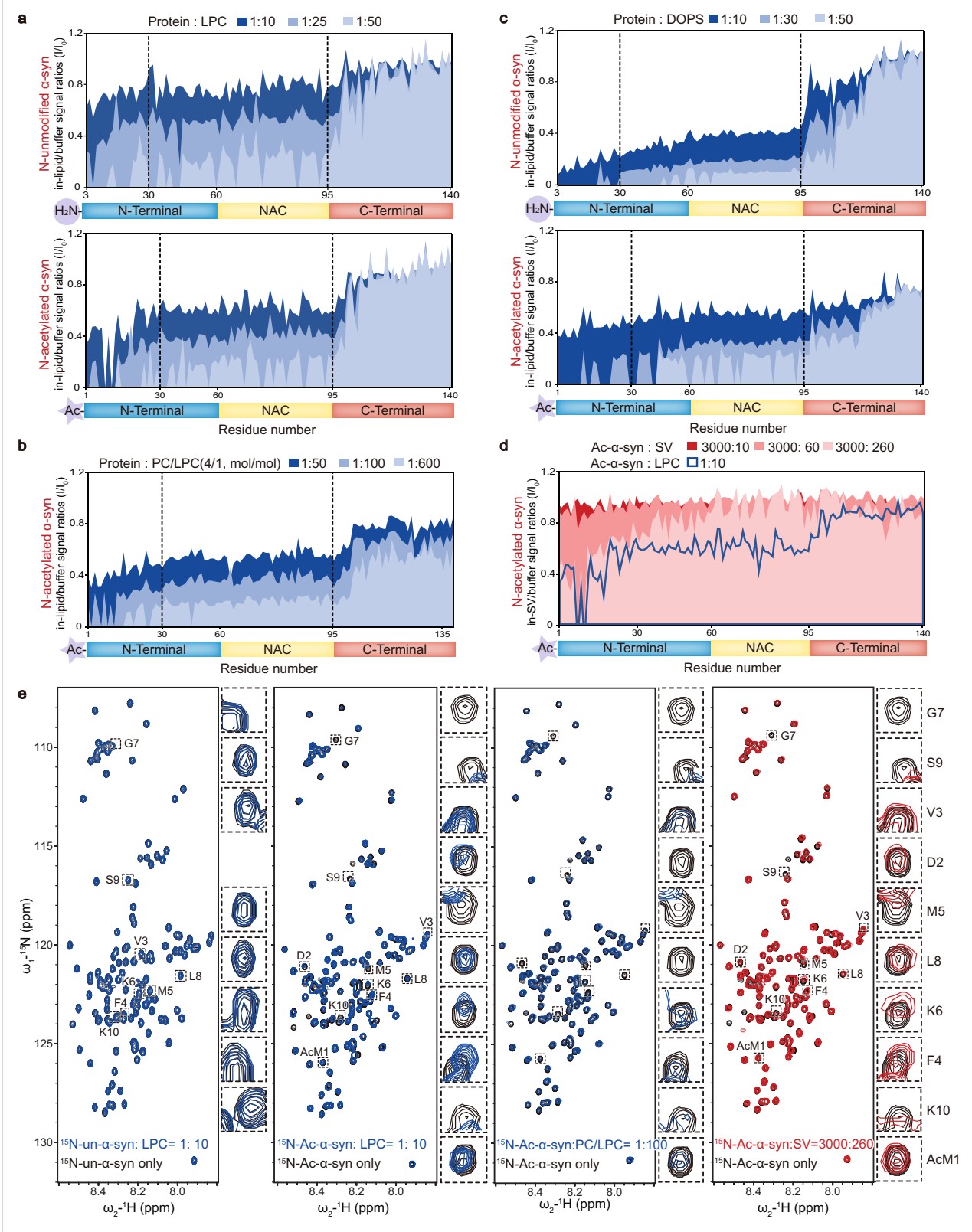

**Figure 3.** N-terminal acetylation increases the α-synuclein (α-syn)–lysophosphatidylcholine (LPC) interaction. Comparisons of residue-resolved nuclear magnetic resonance (NMR) signal intensity ratios (I/I$_0$) of un-α-syn (upper) and Ac-α-syn (lower) during titration with LPC micelles (**a**), LPC-containing liposomes (DOPC:LPC = 4:1, mol:mol) (**b**), and dioleoyl-phosphoserine (DOPS) liposomes (**c**) at indicated protein/lipid molar ratios. Dashed lines highlight the residue positions 30 and 95. (**d**) SVs isolated from mouse brains were employed for NMR titration with $^{15}$N-Ac-α-syn, approximating the

*Figure 3 continued on next page*

*Figure 3 continued*

physiological ratio (α-syn:SV = 4000:300, mol:mol). Residue-resolved NMR signal intensity ratios ($I/I_0$) of Ac-α-syn is titrated by synaptic vesicles (SVs) to that in solution. The molar ratios of SV to Ac-α-syn are indicated. LPC titration in the Ac-α-syn/LPC ratio of 1:10 (blue curve) is overlaid on the SV titration. (**e**) 2D $^1H$-$^{15}N$ HSQC spectra of NMR for un-α-syn with LPC micelles and Ac-α-syn with LPC micelles, LPC-containing liposomes, and mouse SVs. The NMR cross-peaks of the first 10 residues are highlighted and magnified, as depicted on the right side of each spectrum set (Note: the first and second residues of un-α-syn cannot be assigned). *Figure 3—source data 1* (**a–d**): the NMR titration numerical data of α-syn&LPC, Ac-α-syn&LPC, Ac-α-syn&4PC/LPC, α-syn&DOPS, Ac-α-syn&DOPS, and Ac-α-syn&SV.

The online version of this article includes the following source data for figure 3:

**Source data 1.** The NMR titration numerical data.

interactions and their subsequent effects is a critical area of study. Notably, in addition to in-cell NMR findings that demonstrate spontaneous acetylation of α-syn, N-acetylated α-syn has been observed in PD patients' brain tissue, underscoring its physiological relevance (*Anderson et al., 2006*). Our research reveals that N-acetylation enhances the binding of the N-terminal region of α-syn to LPC, thereby facilitating its association with SVs. LPC is a neutral lipid with an inverted-cone shape, known to create high membrane curvature and more lipid packing defects (*Lauwers et al., 2016*). N-acetylation effectively neutralizes the positive charge of α-syn's N-terminus, fostering its insertion into LPC-rich membranes through hydrophobic interactions. Conversely, N-acetylation significantly reduces the affinity of α-syn's N-terminal region for negatively charged PS-containing membranes, where the interaction is mainly electrostatic in nature. Furthermore, compared to DOPS, the interaction between

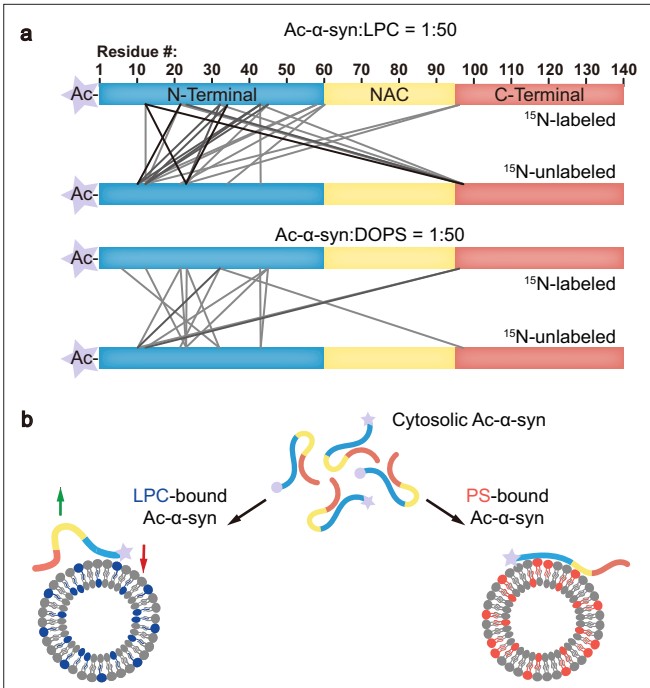

**Figure 4.** Ac-α-syn binding on lysophosphatidylcholine (LPC) shows high intermolecular interactions. (**a**) The cross-linking patterns of Ac-α-syn in the presence of LPC and dioleoyl-phosphoserine (DOPS) mapped by mass spectrometry (MS) at the protein/lipid molar ratio of 1:50. Lines present the inter-molecular cross-linked residues between two individual $^{15}N$-labeled and unlabeled Ac-α-syn. The grayscale of the lines corresponds to the frequency of the cross-linked pairs identified in three individual experiments. Source data are provided in *Supplementary file 1a-f*. (**b**) Ac-α-syn binds strongly to LPC through the N-terminal region (red arrow), and leaves more unbound NAC and C-terminal region for intermolecular interaction (green arrow). In contrast, N-terminal acetylation reduces α-syn's binding to DOPS, and due to the negatively charged headgroup of PS, Ac-α-syn binding on DOPS extends to the NAC region, which limits intermolecular interactions. *Supplementary file 1* (**a-c**) Three biological replicates of identified cross-linked peptides between $^{15}N$-Ac-α-syn and $^{14}N$-Ac-α-syn in LPC; (**d-e**) Three biological replicates of identified cross-linked peptides between $^{15}N$-Ac-α-syn and $^{14}N$-Ac-α-syn in DOPS; Protein:lipid = 1:50, mol:mol.

N-terminally acetylated α-syn and LPC also triggers more intermolecular interactions that support the functional α-syn for SV clustering (*Figure 4b*).

Most recently, we reported that α-syn also interacts with the neutral phospholipid LPC, which is enriched in synaptosomes. LPC can facilitate α-syn to perform its function on clustering SVs (*Lai et al., 2023*). Meanwhile, a recent study showed that LPC can reduce α-syn aggregation (*Zhao et al., 2024*), demonstrating the importance of the α-syn–LPC interaction. Since α-syn is N-terminally acetylated in mammalian cells, the involvement of LPC, not anionic phospholipids, in N-acetylation-enhanced SV clustering demonstrates the importance of our recently reported α-syn–LPC interaction for mediating α-syn's function in SV trafficking. Furthermore, the amount of anionic phospholipids on SVs could change with aging and disease (*Emre et al., 2021*), which may alter the binding modes of α-syn. The α-syn–LPC interaction enhanced by N-acetylation could serve as an alternative mechanism for maintaining the SV clustering function of α-syn.

# Materials and methods

## Key resources table

| Reagent type (species) or resource | Designation | Source or reference | Identifiers | Additional information |
|---|---|---|---|---|
| Strain, strain background (*Escherichia coli*) | BL21(DE3) | BioRad | 156–3003 | Electrocompetent cells |
| Cell line (*Homo-sapiens*) | HEK-293T epithelial-like cells | ATCC | CRL-3216 | Cell line authentication services (STR profiling) are offered by ATCC, not detected mycoplasma contamination |
| Transfected construct (human) | α-syn, Δ30 α-syn to pET22 vector | This paper; *Zhao et al., 2024* | | Constructs saved in C. Liu lab |
| Commercial assay or kit | Neon transfection system kit | Invitrogen | MPK5000 | |
| Chemical compound, drug | 16:0 LPC, DOPC, DOPS, POPC, POPE, biotin-DPPE, cholesterol | Avanti Polar Lipids | 855675, 850375, 840035, 850457, 850757, 870277, 70000 | |
| Chemical compound, drug | Disuccinimidyl suberate (DSS) | Thermo Scientific | 21658 | |
| Chemical compound, drug | uranyl acetate | Sigma Aldrich | CDS021290 | |
| Chemical compound, drug | DiD | Invitrogen | D307 | |
| Software, algorithm | pLink | pLink | V1.9 | For XL-MS data |
| Software, algorithm | SPARKY | SPARKY | V3.115 | For NMR data |
| Software, algorithm | NMRpipe | NMRpipe | Build2018 | For NMR data |
| Software, algorithm | smCamera | TJ Ha's lab | | For single vesicle data |
| Software, algorithm | Dynamics | Wyatt | V7.0 | For DLS data |
| Other | C57BL6 mice | Lingchang Shanghai | 8 wk old, male | |

## Lipids

Lipids used in this study were purchased from Avanti Polar Lipids as follows: 16:0 LPC (1-palmitoyl-2-hydroxy-sn-glycero-3-phosphocholine) (855675), DOPC (1,2-dioleoyl-sn-glycero-3-phosphocholine) (850375), DOPS (1,2-dioleoyl-sn-glycero-3-phospho-L-serine) (840035), POPC (1-palmitoyl-2-oleoyl-glycero-3-phosphocholine) (850457), POPE (1-palmitoyl-2-oleoyl-sn-glycero-3-phosphoethanolamine) (850757), biotin-DPPE (1,2-dipalmitoyl-sn-glycero-3-phosphoethanolamine-N-(cap biotinyl)) (870277), cholesterol (70000).

## Protein purification

N-terminal unmodified human α-syn (un-α-syn) and N-terminal 30-residue truncated α-syn (Δ30 α-syn) were cloned into the pET22 vector. For the N-terminal acetylated α-syn (Ac-α-syn), the fission yeast

NatB complex (N-acetyltransferase) gene was co-expressed with α-syn in bacteria to acetylate its N-terminus (*Johnson et al., 2010*). $^{15}$N isotope-labeled Ac-α-syn or α-syn were produced in M9 minimal medium with uniformly labeled $^{15}$NH$_4$Cl (1 g/L, Cambridge Isotope Laboratories). Not isotope-labeled Ac-α-syn or α-syn were produced in the Luria-Bertani (LB) medium. Expression was induced by 1 mM isopropyl-1-thio-D-galactopyranoside (IPTG) at 37 °C in *E. coli* strain BL21-DE3, which OD$_{600}$ reached 0.8–1.0. The collected bacteria were lysed in 100 mM Tris-HCl (pH 8.0), 1 mM EDTA, and 1 mM phenylmethylsulfonyl fluoride (PMSF). The lysates were boiled for 10 min followed by centrifugation at 16,000 g for 30 min. 20 mg/ml streptomycin was mixed with supernatants. After a 30 min centrifugation at 16,000 g, the pH value of supernatants was adjusted to 3.5 by 2 M HCl, followed by centrifugation at 16,000 g for 30 min. The supernatants were then dialyzed in 25 mM Tris-HCl (pH 8.0) overnight at 4 °C. The dialyzed protein solution was filtered through a 0.22 μm membrane (Merck Millipore, SLGP033RB) and injected onto an ion-exchange Q column (GE Healthcare, 17515601), followed by further purification via size exclusion chromatography (GE Healthcare, Superdex 75).

## Isolation of synaptic vesicles from mouse brains

Neuronal SV were isolated from 8-week-old C57BL6 male mice (Lingchang Shanghai) by following published protocols (*Ahmed et al., 2013*). The experiment was performed at 4 °C or on ice with precooled reagents. Four fresh brains (~0.5 g for each) were homogenized in 25 mL 4 mM HEPES-NaOH (pH 7.4) and 320 mM sucrose buffer (HB) with protease inhibitors by using a PTFE pestle in a 40 mL glass tube (Sigma, P7984). The homogenate was centrifuged at 1,500 g for 10 min. The supernatant (S1) was collected and kept on ice. The pellet was resuspended with 25 mL HB and homogenized, followed by centrifugation at 1500 g for 10 min. Then the supernatant (S2) combined with S1 was centrifuged at 20,000 g for 20 min. The pellet which contains synaptosomes was resuspended with 2 mL HB and then homogenized in 20 mL H$_2$O followed by adding 50 μL of 1 M HEPES-NaOH (pH 7.4) and protease inhibitors. The homogenate was placed on ice for 30 min, and centrifuged at 20,000 g for 20 min. The supernatant was ultra-centrifuged at 70,000 g for 45 min. The pellet was resuspended in buffers indicated in each analysis and homogenized by using a PTFE pestle in a 3 mL glass tube (Sigma, P7734). To further disrupt any remaining SV clusters before experiments, the homogenized SVs were drawn first through a 20-gauge hypodermic needle (Fisher Scientific, 0556121) attached to a 10 ml syringe, and then through a 27-gauge needle (Fisher Scientific, 22557176) and expelled. Mice experiments were conducted according to the protocols approved by the Animal Care Committee of the Interdisciplinary Research Center on Biology and Chemistry (IRCBC), Chinese Academy of Sciences (CAS).

## α-Syn-mediated SV clustering monitored by DLS

The DLS experiments were performed using a DynaPro Nanostar instrument (Wyatt Technology, Santa Barbara, CA). A 50 μL aliquot of each sample containing homogenous SV (total protein concentration of 50 μg/mL) and α-syn variants (3 μM) was used for DLS measurement. The buffer used in this assay was 50 mM HEPES-KOH (pH 7.4), and 150 mM KCl. The following parameters were set for all measurements: correlation function low cutoff was 1.5 μs; correlation function high cutoff was 6×10$^4$ μs; peak radius low cutoff was 0.5 nm; peak radius high cutoff was 10$^4$ nm. The results were reported as the average of 10 consecutive autocorrelation functions (acquisition time = 5 s), and each sample was repeatedly measured three times.

## Negatively stained TEM

An aliquot of 5 μL sample was pipetted onto a glow-discharged carbon-coated copper grid. 5 μL 3% (w/v) uranyl acetate was used for negative staining after washing the grid twice with 5 μL distilled H$_2$O. The grid was air-dried after removing extra uranyl acetate by filter paper. TEM images were obtained by an electron microscope (Thermo Fisher FEI Tecnai G2 TEM) at 120 kV equipped with a LaB$_6$ gun and a FEI BM Eagle CCD camera.

## Liposome preparation

DOPS dissolved in chloroform was evaporated using a dry nitrogen stream. The dried lipid film was hydrated in buffers used for cross-linking (indicated below), and then sonicated in a water bath at 65 °C for 10 min. To mimic the size of synaptic vesicles, the hydrated lipids were extruded 41 times at

65 °C through a poly-carbonate film with a pore size of 50 nm (Whatman Nucleopore Track-Etch) by using an extruder apparatus (Avanti Polar Lipids, 610000). LPC powder was dissolved in buffers used for cross-linking for further analysis.

## Single-vesicle clustering assay

The protein-free liposomes (~100 nm) of POPC, POPE, cholesterol, LPC or DOPS, and biotin-DPPE or DiD (Invitrogen, D307) were prepared in the single-vesicle clustering buffer (25 mM HEPES-NaOH (pH 7.4), 100 mM NaCl) to a total lipid concentration of ~10 mM. The group of unlabeled liposomes contains POPC, POPE, cholesterol, LPC or DOPS, and biotin-DPPE with a molar ratio of (49.9~59.9):20:20:(0~10):0.1. The group of labeled liposomes contain POPC, POPE, cholesterol, LPC or DOPS and DiD with a molar ratio of (49~59):20:20:(0~10):1. LPC or DOPS on both vesicles was varied from 0 to 10%, in expense of POPC.

For the clustering experiments, the unlabeled liposomes (100 µM) were injected into the sample chamber and incubated for 30 min, followed by buffer exchange of 300 µL single-vesicle buffer. 10 µM DiD labeled liposomes were incubated with 20 nM α-syn or Ac-α-syn, and loaded into the sample chamber with 30 min incubation at room temperature. Unbound DiD liposomes and proteins were removed by buffer exchange of 300 µL single-vesicle buffer. Sample slides with multiple channels were monitored in a wide-field TIR fluorescence microscope. Data were acquired with an EMCCD camera, and analyzed with the smCamera program. Six images were taken at random locations within each channel on the quartz slide. The number of liposome interactions (clustering) was determined by counting the number of fluorescent spots from the emission of DiD upon excitation at 633 nm. Details were described previously (*Diao et al., 2013*; *Diao et al., 2012*).

## Delivering $^{15}$N-un-α-syn into mammalian cells by using the electroporation method

Electroporation was performed with a Neon transfection system (Invitrogen, MPK5000). Lyophilized $^{15}$N-un-α-syn was dissolved in Buffer R (Invitrogen, MPK10025) to a final concentration of 300 µM. HEK-293T cells (ATCC, CRL-3216) were collected and washed with PBS for three times to get rid of the medium. Cells were resuspended in $^{15}$N-un-α-syn solution at $8 \times 10^7$ cells per mL. Eight aliquots of 100 µL cell-protein mixture were pipetted and electroporated. Pulse program was: 1400 V pulse voltage, 20 ms pulse width, 1 pulse number. Electroporated cells were added into four 10 cm-diameter dishes with pre-warmed culture medium. Cells were allowed to recover for 5 hr, and were then harvested by trypsinization and washed with PBS four times. Resuspended cells were washed in 160 µL pH-stable L-15 medium (Gibco, 11415064) and 40 µL D$_2$O, and allowed to settle in a 5 mm NMR tube by gentle sedimentation with a hand-driven centrifuge. The supernatant was discarded. Finally, ~500 µL sedimented cell slurry was prepared for NMR measurement.

## In-cell and in vitro solution NMR spectroscopy

In-cell NMR experiments were carried out at 25 °C on a Bruker 900 MHz spectrometer equipped with a cryogenic probe by following a published paper (*Wang et al., 2020*). Bruker standard SOFAST-HMQC pulse sequence (*Schanda and Brutscher, 2005*; *Schanda et al., 2005*) was used and the $^1$H shape pulse efficiency was optimized for collecting the 2D NMR spectrum of the in-cell samples with 80 scans. The delay time (D1) was set to 0.29 s, and 1024 and 128 complex points were used for $^1$H and $^{15}$N, respectively. The experiment duration time is 61 min 16 s. Cell viability before and after the SOFAST-HMQC NMR experiment was assessed by trypan blue staining. Cell viability remained above 90% after the NMR experiments.

In vitro NMR experiments were carried out at 25 °C on an 800 MHz Agilent spectrometer equipped with a cryogenic probe (Agilent Technologies). The buffer used in all NMR titration assays was 50 mM sodium phosphate buffer (pH 6.5) containing 50 mM NaCl and 10% D$_2$O (v/v). All of the LPC or liposomes were prepared with a total molar concentration of 50 mM, and gradually added to the 500 µL 50 µM N-acetylated or unmodified α-syn in a series of NMR titration assays. In the NMR SV-titration experiment, each spectrum was acquired from separately prepared samples with increasing amounts of SV in 50 µM N-acetylated α-syn. The molar concentration of SVs was calculated according to parameters from a published paper (*Takamori et al., 2006*).

All the NMR data were processed using NMRPipe and analyzed by SPARKY (*Delaglio et al., 1995*). Backbone resonance assignment of N-acetylated and unmodified α-syn was accomplished according to a published paper (*Theillet et al., 2016*) and a Biological Magnetic Resonance Bank (BMRB) entry 6968 (*Bermel et al., 2006*), respectively.

### α-Syn multimer cross-linking experiment

LPC micelles, DOPS liposomes, and α-syn variants were mixed in a cross-linking-buffer (10 mM HEPES-NaOH, pH 7.4). The stock of cross-linker disuccinimidyl suberate (DSS) (Thermo Scientific, 21658) was dissolved in DMSO to 40 mM. 50 μM protein and 2 mM DSS were incubated at indicated concentrations of lipid in the cross-linking buffer at 25°C for 1 hr. Crosslinking was terminated by the addition of 100 mM Tris-HCl (pH 7.4) at 25°C for 10 min. For identification of the inter-molecular cross-linked peptide pairs from α-syn multimers by MS analysis, the half amount of α-syn was $^{15}$N-labeled α-syn.

### Protein cross-linking mass spectrometry (XL-MS) analysis

The cross-linked samples were precipitated using acetone at −20 °C for 30 min. Precipitates were dried in air and resuspended in 8 M urea in 100 mM Tris-HCl (pH 8.5). Then, 5 mM TCEP was adding for 20 min incubation, followed by adding 10 mM iodoacetamide for 15 min for an alkylation reaction in the dark. Then, the samples were diluted to 2 M urea in 100 mM Tris-HCl (pH 8.5), and digested with trypsin (molar ratio of protein to trypsin was 50:1) at 37 °C for 16 hr in the presence of 1 mM $CaCl_2$ and 20 mM methylamine. Digestion was terminated by adding formic acid to 5% final concentration (v/v). After desalting by C18 desalting tips, the digested samples were air-dried and stored at −80°C until analysis.

The digested peptides were analyzed by online nanoflow liquid chromatography-tandem mass spectrometry (LC−MS/MS). Briefly, nano LC−MS/MS experiments were performed on an EASY-nLC 1000 system (Thermo Scientific) connected to an Orbitrap Q Exactive HF (Thermo Scientific) through a nanoelectrospray ion source. The peptides were separated on a nano-column (C18, 100 μm×15 cm, 1.9 μm, 120 Å) and further analyzed using an Orbitrap Q Exactive HF mass spectrometer. A 60 min gradient was used in the nano LC at 300 nL/min flow rate: 0–4 min from 4% buffer B (80% ACN, 0.1% FA), 96% buffer A (0.1% FA in water) to 8% buffer B, 4–45 min from 8% buffer B to 22% buffer B, 45–53 min from 22% buffer B to 30% buffer B, then increasing to 95% buffer B in the next 41 min, and stay 95% buffer B for 3 min. One full-scan mass spectrum (350–1500 m/z) at a resolution of 60,000 followed by HCD fragmentation and detection of the fragment ions (scan range from 350 to 1500 m/z) in Orbitrap at a 27% normalized collision energy was repeated continuously.

The cross-linked peptide pairs were identified by pLink (*Fan et al., 2015*) first. The pLink search parameters used in this study include: (1) 20 ppm for a window of precursor mass tolerance. (2) 4 ppm for a window of fragment mass tolerance. (3) Lys for cross-linking sites of the cross-linker DSS. (4) 138.0680796 Da for xlink mass shift. (5) 156.0786442 Da for monolink mass shift. (6) 57.02146 Da for fixed carbamidomethyl-Cys modification mass shift. (7) 15.99491 Da for variable Met modification mass shift. (8) 4–100 amino acids per chain for peptide length. (9) 400–10,000 Da per chain for peptide mass. (9) 2 missed cleavage sites per chain for trypsin digestion of protein. The search results were filtered with a false rate of less than 5% and E-values of less than 0.0001.

The pLink readout data were filtered with a score less than $10^{-6}$ and mass error for crosslinking assignments less than 4 ppm. To identify the inter-protein cross-linked peptide pairs, the spectrum (ΔRT <0.5 s) of the two cross-linked peptide segments needs to be manually checked with four peaks of $^{14}$N-peptide and $^{14}$N-peptide, $^{14}$N-peptide and $^{15}$N-peptide, $^{15}$N-peptide and $^{14}$N-peptide and $^{15}$N-peptide and $^{15}$N-peptide.

### Cell Line

HEK-293T cells were obtained from the American Type Culture Collection (CRL-3216).

### Animal statement

C57BL6 male mice were from Lingchang Shanghai. All animal procedures were performed in accordance with the protocols approved by the Animal Care Committee of the Interdisciplinary Research Center on Biology and Chemistry, Chinese Academy of Sciences (mouse protocol number 20230110005).

## Data availability

All data are available in the main text or the extended data. Source data are provided in this paper. The XL-MS raw data and pLink searched data have been uploaded to iProX with the accession number IPX0009772000. Any data supporting the findings of this manuscript are available from the corresponding author upon reasonable request.

## Acknowledgements

CL and DL thank the support from the National Key R&D Program of China (2019YFE0120600 to CL), National Natural Science Foundation (NSF) of China (92353302 and 32170683 to DL; 82188101 and 32171236 to CL), the Science and Technology Commission of Shanghai Municipality (STCSM) (Grant No. 22JC1410400 to CL), the Shanghai Pilot Program for Basic Research – Chinese Academy of Science, Shanghai Branch (Grant No. CYJ-SHFY-2022–005 to CL), the CAS Project for Young Scientists in Basic Research (Grant No.YSBR-095 to CL), Shanghai Basic Research Pioneer Project to CL, the Strategic Priority Research Program of the Chinese Academy of Sciences (Grant No. XDB1060000 to CL). This work was supported by the National Institutes of Health (R01NS102181, R01NS113960, 1RF1NS126342 and R21NS127939 to JB), and the Michael J Fox Foundation (16661 to JD and 16164 to JB). We thank the staff members of the Nuclear Magnetic Resonance System(https://cstr.cn/31129. 02.NFPS.NMRSystem) at the National Facility for Protein Science in Shanghai (https://cstr.cn/31129. 02.NFPS), for providing technical support and assistance in data collection and analysis.

## Additional information

### Funding

| Funder | Grant reference number | Author |
|---|---|---|
| National Key R&D Program of China | 2019YFE0120600 | Cong Liu |
| National Natural Science Foundation of China | 92353302 | Dan Li |
| National Natural Science Foundation of China | 32170683 | Dan Li |
| National Natural Science Foundation of China | 82188101 | Cong Liu |
| National Natural Science Foundation of China | 32171236 | Cong Liu |
| Shanghai Municipal Science and Technology Commission | 22JC1410400 | Cong Liu |
| Chinese Academy of Sciences, Shanghai Branch | CYJ-SHFY-2022–005 | Cong Liu |
| Chinese Academy of Sciences | YSBR-095 | Cong Liu |
| Shanghai Basic Research Pioneer Project | | Cong Liu |
| Strategic Priority Research Program of the Chinese Academy of Sciences | XDB1060000 | Cong Liu |
| National Institutes of Health | R01NS102181 | Jacqueline Burré |
| National Institutes of Health | R01NS113960 | Jacqueline Burré |
| National Institutes of Health | 1RF1NS126342 | Jacqueline Burré |

| Funder | Grant reference number | Author |
| --- | --- | --- |
| National Institutes of Health | R21NS127939 | Jacqueline Burré |
| Michael J Fox Foundation for Parkinson's Disease Research | 16661 | Jiajia Diao |
| Michael J Fox Foundation for Parkinson's Disease Research | 16164 | Jacqueline Burré |
| National Institutes of Health | R01NS121077 | Jacqueline Burré Jiajia Diao |

The funders had no role in study design, data collection and interpretation, or the decision to submit the work for publication.

### Author contributions

Chuchu Wang, Data curation, Investigation, Writing - original draft, Writing - review and editing; Chunyu Zhao, Data curation, Formal analysis, Investigation, Visualization, Writing - original draft; Hu Xiao, Data curation, Validation; Jiali Qiang, Zhenying Liu, Jinge Gu, Data curation, Investigation; Shengnan Zhang, Yaoyang Zhang, Investigation; Dan Li, Investigation, Writing - review and editing; Jacqueline Burré, Writing - review and editing; Jiajia Diao, Conceptualization, Resources, Writing - original draft, Project administration, Writing - review and editing; Cong Liu, Conceptualization, Resources, Supervision, Writing - original draft, Project administration, Writing - review and editing

### Author ORCIDs

Chuchu Wang http://orcid.org/0000-0003-2015-7331
Chunyu Zhao https://orcid.org/0000-0003-0168-2130
Dan Li https://orcid.org/0000-0002-1609-1539
Yaoyang Zhang https://orcid.org/0000-0001-5363-9834
Jacqueline Burré https://orcid.org/0000-0001-8968-248X
Jiajia Diao https://orcid.org/0000-0003-4288-3203
Cong Liu https://orcid.org/0000-0003-3425-6672

### Ethics

All animal procedures were performed in accordance with the protocols approved by the Animal Care Committee of the Interdisciplinary Research Center on Biology and Chemistry, Chinese Academy of Sciences (mouse protocol number 20230110005).

Reviewer #1 (Public review): https://doi.org/10.7554/eLife.97228.3.sa1
Reviewer #2 (Public review): https://doi.org/10.7554/eLife.97228.3.sa2
Author response https://doi.org/10.7554/eLife.97228.3.sa3

## Additional files

### Supplementary files

Supplementary file 1. Supplementary tables.

MDAR checklist

### Data availability

Figure 1-source data 1, Figure 3-source data 2, Supplementary file 1a-f for Figure 4a contain the numerical data used to generate the figures. The XL-MS raw data and pLink searched data have been uploaded to iProX with the accession number IPX0009772000.

The following dataset was generated:

| Author(s) | Year | Dataset title | Dataset URL | Database and Identifier |
|---|---|---|---|---|
| Wang S, Zhao C, Xiao H, Qiang J, Liu Z, Gu J, Zhang S, Li D, Zhang Y, Burré J, Diao J, Liu C | 2024 | N-acetylation of α-synuclein enhances synaptic vesicle clustering mediated by α-synuclein and lysophosphatidylcholine | https://www.iprox.cn/page/project.html?id=IPX0009772000 | iProx, IPX0009772000 |

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
