## [Editor Report · eLife assessment]

In this **useful** study, the authors show that N-acetylation of synuclein increases clustering of synaptic vesicles in vitro and that this effect is mediated by enhanced interaction with lysophosphatidylcholine. While the evidence for enhanced clustering is largely **solid**, the biological significance remains unclear.

---

## [Referee Report · Reviewer #1 (Public review)]

⍺-synuclein (syn) is a critical protein involved in many aspects of human health and disease. Previous studies have demonstrated that post-translational modifications (PTMs) play an important role in regulating the structural dynamics of syn. However, how post-translational modifications regulate syn function remains unclear. In this manuscript, Wang et al. reported an exciting discovery that N-acetylation of syn enhances the clustering of synaptic vesicles (SVs) through its interaction with lysophosphatidylcholine (LPC). Using an array of biochemical reconstitution, single vesicle imaging, and structural approaches, the authors uncovered that N-acetylation caused distinct oligomerization of syn in the presence of LPC, which is directly related to the level of SV clustering. This work provides novel insights into the regulation of synaptic transmission by syn and might also shed light on new ways to control neurological disorders caused by syn mutations.

---

## [Referee Report · Reviewer #2 (Public review)]

Summary:

In this manuscript, the authors provide evidence that posttranslational modification of synuclein by N-acetylation increases clustering of synaptic vesicles in vitro. When using liposomes the authors found that while clustering is enhanced by the presence of either lysophosphatidylcholine (LPC) or phosphatidylcholine in the membrane, N-acetylation enhanced clustering only in the presence of LPC. Enhancement of binding was also observed when LPC micelles were used, which was corroborated by increased intra/intermolecular cross-linking of N-acetylated synuclein in the presence of LPC.

Strengths:

It is known for many years that synuclein binds to synaptic vesicles but the physiological role of this interaction is still debated. The strength of this manuscript is clearly in the structural characterization of the interaction of synuclein and lipids (involving NMR-spectroscopy) showing that the N-terminal 100 residues of synuclein are involved in LPC-interaction, and the demonstration that N-acetylation enhances the interaction between synuclein and LPC.

Weaknesses:

Lysophosphatides form detergent-like micelles that destabilize membranes, with their steady-state concentrations in native membranes generally being a lot lower than in the experiments reported here. Since no difference in binding between the N-acetylated and unmodified form was observed when the acidic phospholipid phosphatidylserine was included. It remains unclear to which extent binding to LPC is physiologically relevant, particularly in the light of recent reports from other laboratories showing that synuclein may interact with liquid-liquid phases of synapsin I, or associate with the unfolded regions of VAMP that both were reported to cause vesicle clustering.

---

## [Author Response]

The following is the authors’ response to the original reviews.

**Reviewer #1 (Public Review):**
⍺-synuclein (syn) is a critical protein involved in many aspects of human health and disease. Previous studies have demonstrated that post-translational modifications (PTMs) play an important role in regulating the structural dynamics of syn. However, how post-translational modifications regulate syn function remains unclear. In this manuscript, Wang et al. reported an exciting discovery that N-acetylation of syn enhances the clustering of synaptic vesicles (SVs) through its interaction with lysophosphatidylcholine (LPC). Using an array of biochemical reconstitution, single vesicle imaging, and structural approaches, the authors uncovered that N-acetylation caused distinct oligomerization of syn in the presence of LPC, which is directly related to the level of SV clustering. This work provides novel insights into the regulation of synaptic transmission by syn and might also shed light on new ways to control neurological disorders caused by syn mutations.

We thank the reviewer for appreciating the importance of our work and his/her positive comments.

**Reviewer #1** (**Recommendations For The Authors):**(1) The authors employed DLS to quantify the percentage of SV clustering in Fig. 1c and d. As DLS usually measures particle size distribution, I am not sure how the data was plotted in Fig. 1c and d. It would be great to show a representative raw dataset here.

We thank the reviewer for the comment. To address this, we have put four representative DLS datasets of different α-Syn variants mediating SV clustering for clarification (Author response image 1). Rather than presenting the particle distribution based on the light scattering intensity, DLS can also convert the intensity to present the data as particle size distribution based on the particle number counts. In our analysis, particle diameters around 50 nm are considered to represent single SV species, whereas diameters larger than 120 nm indicate SV clusters. Specifically, as shown in Author response image 1, adding Ac-α-syn to a homogeneous SV sample altered the distribution from one single SV particle species (Author response image 1d) to three distinct species (Author response image 1a); this resulted in 68.5% of the particles being single SVs and 31.5% being SV clusters.

**Author response image 1. sa3fig1:** Representative raw dataset of α-Syn-mediated synaptic vesicle (SV) clustering monitored by dynamic light scattering (DLS). The gray-colored rows represent small particles (< 5 nm) that contributed zero to the particle number count.

(2) Syn-lipid interactions are known to be altered by mutations involved in neurodegenerative diseases. I am wondering how those mutations will affect SV clustering mediated by the interaction of LPC with N-acetylated syn.

We thank the reviewer for the insightful comment. Our data indicate that N-acetylation enhances the binding of the N-terminal region of α-syn to LPC, thereby facilitating SV clustering. This enhancement benefits from the fact that N-acetylation effectively neutralizes the positive charge of α-syn’s N-terminal region, promoting its insertion into LPC-rich membranes through hydrophobic interactions. Therefore, we envision that any mutation that weakens membrane binding capability of the N-terminal unmodified α-Syn may decrease SV clustering mediated by the interaction between the Ac-α-syn and LPC.

In a separated work (*doi: 10.1093/nsr/nwae182*, Fig. S8), we compared the binding affinity of LPC with wild-type N-terminal un-modified α-syn and six Parkinson’s disease (PD) familial mutants (A30P, E46K, H50Q, G51D, A53E, and A53T). Among these, only the A30P mutation showed a significant decrease in binding with LPC. Furthermore, using the same single vesicle assay setup, in another paper (*doi: 10.1073/pnas.2310174120*, Fig. 4C), we demonstrated that the A30P-mutated α-Syn lost its ability to facilitate SV clusters. Therefore, among the six PD mutations, the A30P mutation may significantly impact the SV clustering mediated by Ac-α-syn LPC interaction.

(3) The crosslinking data in Fig. 4 was obtained using LPC or PS liposomes. I am wondering if these results truly mimic physiological conditions. Could the authors use SVs for these experiments?

We thank the reviewer for the suggestion. To elucidate the mechanistic differences between N-terminal unmodified α-syn and N-acetylated α-syn, we utilized pure LPC and PS liposomes for clarity. If using natural source SVs, which contain many synaptic proteins, could complicate or obscure the interaction patterns of Ac-α-syn due to potential crosstalk with other SV proteins. Additionally, the complex lipid environment of SV membranes would not help us decipher the specific molecular mechanism by which Ac-α-Syn facilitates SV clustering through LPC.

**Reviewer #2 (Public Review):**
Summary:In this manuscript, the authors provide evidence that posttranslational modification of synuclein by N-acetylation increases clustering of synaptic vesicles in vitro. When using liposomes the authors found that while clustering is enhanced by the presence of either lysophosphatidylcholine (LPC) or phosphatidylcholine in the membrane, N-acetylation enhanced clustering only in the presence of LPC. Enhancement of binding was also observed when LPC micelles were used, which was corroborated by increased intra/intermolecular cross-linking of N-acetylated synuclein in the presence of LPC.Strengths:It is known for many years that synuclein binds to synaptic vesicles but the physiological role of this interaction is still debated. The strength of this manuscript is clearly in the structural characterization of the interaction of synuclein and lipids (involving NMR-spectroscopy) showing that the N-terminal 100 residues of synuclein are involved in LPC-interaction, and the demonstration that N-acetylation enhances the interaction between synuclein and LPC.

We thank the reviewer for their positive assessment of our work.

Weaknesses:Lysophosphatides form detergent-like micelles that destabilize membranes, with their steady-state concentrations in native membranes being low, questioning the significance of the findings. Oddly, no difference in binding between the N-acetylated and unmodified form was observed when the acidic phospholipid phosphatidylserine was included. It remains unclear to which extent binding to LPC is physiologically relevant, particularly in the light of recent reports from other laboratories showing that synuclein may interact with liquid-liquid phases of synapsin I that were reported to cause vesicle clustering.

We appreciate the reviewers’ insightful comments. Indeed, in another paper (*doi: 10.1093/nr/nwae182),* employing conventional α-Syn pull-down assay and LC-MS lipidomics method, we found that α-Syn has a preference for binding to lysophospholipids across *in vivo* and *in vitro* systems. Additionally, by comparing the lipid compositions of mouse brains, SVs and SV lipid-raft membranes, we found LPC levels to be twice as high in SVs compared to brain homogenates, and twice as high in lipid-raft membranes compared to non-lipid-raft membranes. Altogether, these findings emphasize the physiological relevance of understanding the mechanism by which Ac-α-syn mediated SV clustering through LPC.

Liquid-liquid phase separation has been implicated in the assembly and maintenance of SV clusters, and we believe that the SV cluster liquid phase is interconnected by highly abundant proteins with multivalent low-affinity interactions. Besides the previously discovered protein-protein interactions between α-Syn and synapsin (*doi: 10.1016/j.jmb.2021.166961*) or VAMP2 (*doi: 10.1038/s41556-024-01456-1*) that contribute to SV condensates, protein-lipid interactions between α-Syn and acidic phospholipids or LPC may also play a role. Furthermore, post-translational modifications, such as N-acetylation of α-Syn, may also contribute to SV condensates.

**Reviewer #2 (Recommendations For The Authors):**
In Fig. 2, the authors indicate that for the binding assay both vesicle populations, the immobilized "acceptor" and the superfused "donor" population were labeled with different fluorescent dyes whereas in the text it is stated that the immobilized acceptor liposomes were unlabeled. Please clarify. Moreover, a control is missing showing that binding indeed depends on the immobilised liposome fraction and does not occur in their absence. This control is important because due to the long incubation times non-specific adsorption may occur which may be enhanced by adding destabilizing LPC or charged PS to the membrane.

We thank the reviewer for pointing out this inconsistency. To avoid signal leakage from a high concentration of DiD vesicles upon green laser irradiation, we immobilized unlabeled vesicles. We have revised the Figure 2a as well as the figure caption.

Regarding the control mentioned by the reviewer, we agree with the reviewer that non-specific binding could occur with the long incubation. In fact, a layer of highly dense liposomes (100 μM) immobilized on the imaging surface is also for reducing non-specific interactions. In the absence of this layer of immobilized liposomes, we did see a high level of non-specific binding that significantly impacted our experiments. Therefore, we need to perform clustering experiments in the presence of immobilized liposomes.